# Therapeutic Potential of Endothelial Colony-Forming Cells in Ischemic Disease: Strategies to Improve their Regenerative Efficacy

**DOI:** 10.3390/ijms21197406

**Published:** 2020-10-07

**Authors:** Pawan Faris, Sharon Negri, Angelica Perna, Vittorio Rosti, Germano Guerra, Francesco Moccia

**Affiliations:** 1Laboratory of General Physiology, Department of Biology and Biotechnology “L. Spallanzani”, University of Pavia, 27100 Pavia, Italy; faris.pawan@unipv.it (P.F.); sharon.negri01@universitadipavia.it (S.N.); 2Department of Medicine and Health Sciences “Vincenzo Tiberio”, University of Molise, 86100 Campobasso, Italy; angelicaperna@gmail.com (A.P.); germano.guerra@unimol.it (G.G.); 3Center for the Study of Myelofibrosis, Laboratory of Biochemistry, Biotechnology and Advanced Diagnosis, IRCCS Policlinico San Matteo Foundation, 27100 Pavia, Italy; v.rosti@smatteo.pv.it

**Keywords:** cardiovascular disease, ischemic disorders, therapeutic angiogenesis, endothelial colony forming cells, signaling pathways, pharmacological conditioning, genetic modification

## Abstract

Cardiovascular disease (CVD) comprises a range of major clinical cardiac and circulatory diseases, which produce immense health and economic burdens worldwide. Currently, vascular regenerative surgery represents the most employed therapeutic option to treat ischemic disorders, even though not all the patients are amenable to surgical revascularization. Therefore, more efficient therapeutic approaches are urgently required to promote neovascularization. Therapeutic angiogenesis represents an emerging strategy that aims at reconstructing the damaged vascular network by stimulating local angiogenesis and/or promoting de novo blood vessel formation according to a process known as vasculogenesis. In turn, circulating endothelial colony-forming cells (ECFCs) represent truly endothelial precursors, which display high clonogenic potential and have the documented ability to originate de novo blood vessels in vivo. Therefore, ECFCs are regarded as the most promising cellular candidate to promote therapeutic angiogenesis in patients suffering from CVD. The current briefly summarizes the available information about the origin and characterization of ECFCs and then widely illustrates the preclinical studies that assessed their regenerative efficacy in a variety of ischemic disorders, including acute myocardial infarction, peripheral artery disease, ischemic brain disease, and retinopathy. Then, we describe the most common pharmacological, genetic, and epigenetic strategies employed to enhance the vasoreparative potential of autologous ECFCs by manipulating crucial pro-angiogenic signaling pathways, e.g., extracellular-signal regulated kinase/Akt, phosphoinositide 3-kinase, and Ca^2+^ signaling. We conclude by discussing the possibility of targeting circulating ECFCs to rescue their dysfunctional phenotype and promote neovascularization in the presence of CVD.

## 1. Introduction

Cardiovascular disease (CVD) comprises a group of heart and circulatory disorders, which are regarded as a global medical and economic issue with high prevalence and mortality rates [1]. The World Health Organization (WHO) and Global Burden Disease (GBD) have listed CVD as the first cause of death worldwide [2]. It was estimated that 17.9 million people died from CVD in 2016, representing 31% of all global deaths. Of these deaths, 85% were due to heart attack and stroke [1]. In line with these observations, ischemic heart disease emerged as the main contributor to disease burden as assessed by the evaluation of disability-adjusted life years [3]. CVD includes aortic atherosclerosis, coronary artery disease (CAD), which can ultimately lead to acute myocardial infarction (AMI), stroke, and peripheral arterial disease (PAD) [4]. CVD is characterized by the narrowing or occlusion of specific vascular beds, e.g., coronary, brain, or skeletal muscle, which are caused by endothelial dysfunction [4]. Vascular regenerative surgery represents the most currently employed therapeutic option to treat ischemic disorders and re-establish tissue perfusion [5]. Unfortunately, not all the patients are amenable to surgical revascularization through coronary artery bypass surgery, percutaneous coronary intervention, or the deployment of intracoronary stents [5]. Pharmacological treatment with a wide array of drugs, including statins, prostanoids, and phosphodiesterase inhibitors, can be exploited as an adjuvant therapy to alleviate the symptoms and burden of PAD when surgical intervention is not feasible or fails to restore blood flow [6]. Therefore, novel and more efficient therapeutic approaches to promote neovascularization and rescue blood supply to ischemic tissues are urgently required.

Therapeutic angiogenesis represents an emerging strategy to reconstruct the damaged vascular network by stimulating local angiogenesis and/or promoting de novo blood vessel formation according to a process known as vasculogenesis. Current strategies to induce vascular regrowth of ischemic tissues include the delivery of pro-angiogenic genes or peptides, e.g., vascular endothelial growth factor (VEGF)-A and fibroblast growth factor (FGF)-4 [5], or stem cell transplantation [7]. Cell-based therapy consists of the mobilization or transplantation of multiple types of pro-angiogenic stem cells, including bone marrow-derived mesenchymal stem cells (MSCs), hematopoietic cells, and endothelial progenitor cells (EPCs) [6,7,8,9]. As vascular endothelial cells possess limited regenerative capacity, there is growing interest in circulating EPCs due to their recognized role in the maintenance of endothelial integrity, function, and postnatal neovascularization [10,11,12,13]. EPCs were originally identified as a specific population of bone marrow-derived mononuclear cells (MNCs), which were mobilized upon an ischemic insult and postulated to promote de novo blood formation also in adult organisms [14]. This landmark discovery fostered an intense search for the most effective strategy to utilize EPCs for the regenerative therapy of ischemic disorders. However, the therapeutic use of EPCs has been hampered by inconsistent definitions and different protocols employed to isolate and expand them from peripheral and umbilical cord blood [15,16,17]. It has been demonstrated that two distinct and well-defined EPC subtypes may emerge from cultured mononuclear cells, which differ in their ontology and reparative mechanisms. These EPC subtypes include myeloid angiogenic cells (MACs), also termed as circulating angiogenic cells (CACs), pro-angiogenic hematopoietic cells [1], pro-angiogenic circulating hematopoietic stem/progenitor cells (pro-CHSPCs or pro-CPCs), or “early” EPCs, and endothelial colony-forming cells (ECFCs). MACs originate from the myeloid–monocytic lineage and support endothelial repair and vascular regeneration through largely paracrine signals [18]. In turn, ECFCs represent the only known truly endothelial precursor, as they lack hematopoietic markers, display high clonogenic potential, are able to aggregate into bidimensional tube networks in vitro and to originate patent vessels in vivo [19,20]. Therefore, ECFCs provide the most promising cellular candidate for therapeutic purposes by physically engrafting within injured endothelial and also delivering pro-angiogenic cues to adjacent endothelial cells [21]. Understanding the signal transduction pathways that drive their angiogenic behavior is predicted to remarkably improve the therapeutic outcome of ECFCs when they will be probed in clinical trials on human subjects.

The current review provides a brief outline on the origin and characterization of ECFCs and a summary of the progress in preclinical studies aiming at assessing their efficacy in a variety of ischemic disorders, including AMI, PAD, ischemic brain disease, and retinopathy (Figure 1). Then, we describe how to increase ECFCs’ vasoreparative potential by boosting specific pro-angiogenic signaling pathways through pharmacological conditioning and gene manipulation (Figure 1).

## 2. Endothelial Colony-Forming Cells: Origin, Mechanisms of Actions, and Evidence for Therapeutic Applications

### 2.1. Endothelial Colony-Forming Cells (ECFCs): Origin, Characterization and Biological Activity

ECFCs, also referred to as late EPCs, late outgrowth EPCs, blood outgrowth endothelial cells, and endothelial outgrowth cells, are the only EPC subset faithfully belonging to the endothelial lineage [16]. The existence in the peripheral blood of a sub-population of MNCs that are home to the ischemic site and promote neovascularization, thereby rescuing local blood perfusion, which led Isner’s group to introduce the concept of EPC [14]. Nevertheless, a recent consensus of nomenclature proposed abandoning the term EPC in favor of a more stringent definition based upon the immunophenotypical and functional features of the cellular population under investigation [16]. ECFCs are isolated by plating the MNC fraction of peripheral or umbilical cord blood on type 1 rat collagen-coated dishes, and they emerge as adherent cells after 2–3 weeks in culture under endothelial conditions [22,23]. ECFC colonies display a cobblestone-like endothelial cell morphology and, although defining a panel of specific cell surface antigens remains an elusive task, ECFCs present typical endothelial cell markers (e.g., CD31, CD34, CD146, CD309, CD144, VEGF receptor 2 (VEGFR-2), VE-cadherin, and von Willebrand factor), while they lack hematopoietic (e.g., CD14, CD45, and CD115) as well as mesenchymal stem cell (MSC) (CD70 and α-smooth muscle actin) antigens [24,25]. Furthermore, ECFCs present the cytosolic, but not membrane, expression of the stemness marker, CD133 [26]. ECFCs show clonal proliferative potential, assembly into bidimensional capillary-like networks in Matrigel scaffolds, and integrate within host vasculature in vivo [22,23,27,28,29]. As previously observed in hematopoietic stem cells, individual umbilical cord blood (UCB)-derived ECFCs generate colonies with a defined hierarchy of proliferative potentials, including high proliferative potential (HPP)-ECFCs (HPP-ECFCs), which may be replated in secondary and tertiary colonies and retain high telomerase activity, low proliferative potential (LPP)-ECFCs, ECFC clusters, and non-proliferating endothelial cells [22]. Furthermore, the regenerative potential of UCB-derived HPP-ECFCs is sensitive to high cytosolic aldehyde dehydrogenase (ALD^hi^) activity [30], which is likely to protect ECFCs from oxidative damage. In addition, highly proliferative ECFC clones were shown to be genomically stable until very late passages (at least 20−22) [31], when they are still able to assembly into capillary-like structures in vitro and form patent vessels in vivo [32].

The origin of circulating ECFCs remains to be fully clarified [33]. It has originally been suggested that ECFCs are mobilized from bone marrow [34], which also represents the source of MACs. Nevertheless, a recent investigation by Fujisawa and co-workers demonstrated that ECFCs isolated from the peripheral blood and venous wall of male patients previously transplanted with female bone marrow exhibited an XY genotype [35]. Conversely, CD45^+^ MACs did not express the endothelial marker CD31 and had an XX genotype, which was consistent with their bone marrow origin [35]. This finding lent further credit to the notion that ECFCs colonies do not emerge from bone marrow-derived MNCs [36]. Recent investigations demonstrated that ECFCs could arise from multiple progenitor cell niches located within the vascular wall of large vessels, including aorta [23], lung tissue [37], pulmonary artery [38], saphenous vein [39], and placenta [40], or, they can be obtained from white-adipose tissue [41] or from human-induced pluripotent stem cells (hiPSCs) sorted for Neuropilin 1 and CD31 [42]. RNA sequencing confirmed that the gene expression profile of circulating ECFCs was closer to human coronary artery endothelial cells and human umbilical vein endothelial cells as compared to a non-endothelial cell type, such as adipose tissue-derived stromal vascular fraction [43]. Interestingly, the transcriptomic profile of ECFCs bears more resemblance with that of microvascular, rather than macrovascular, endothelial cells [44]. Resident vascular endothelial stem cells (VESCs), displaying clonogenic potential and able to induce endothelial regeneration upon large vessel injury, were recently identified throughout peripheral circulation in mice [33,45]. VESCs were identified by the expression of several cell surface markers, including Peg3/PW1 [46], Protein C receptor (Procr, CD201) [47], and c-Kit (CD117). In addition, a quiescent VESC population was identified by using flow cytometry to carry out the side population (SP) assay [48], which consists of measuring the efflux rate of a fluorescent dye, such as Hoechst 33342. Stem cells may indeed efflux Hoechst 33,342 at a faster rate as compared to differentiated cells [49] due to the higher expression of a novel stem cell marker, the ABC-binding cassette transporter ABCG2. These endothelial–SP cells were dispersed along the vascular tree, showed robust proliferative potential, and were able to promote neovascularization and rescue blood perfusion in mouse models of hindlimb ischemia [47]. It is unclear whether circulating ECFCs are contributed by VESCs sloughed off from the vascular wall as well, as it remains to be fully established that they are mobilized from the vascular niches following an ischemic insult. Nevertheless, there is no doubt that ECFC transplantation promotes neovascularization and rescues blood perfusion in ischemic tissues [50].

ECFCs stimulate vascular repair by exploiting different mechanisms, which may cooperate to favor tissue revascularization: (1) physical engraftment within emerging neovessels [27,28]; (2) paracrine release of pro-angiogenic mediators, which stimulate sprouting angiogenesis from adjacent capillaries, and chemotactic cues [51,52] that recruit mural and/or inflammatory cells; [53] (3) secretion of microvesicles and/or exosomes, which underlie ECFCs-dependent vascular repair in multiple pathological settings [54,55,56]; and (4) support of the regenerative potential of MSCs [55,57] and adipose stromal cells [58]. Furthermore, ECFCs may efficiently interact with MACs during the neovascularization process. As discussed in [59,60], circulating MACs are the first EPC subtype to be recruited within the hypoxic tissue, thereby releasing a host of growth factors that recruit ECFCs from peripheral blood or the vascular progenitor cell niches. Next, ECFCs proliferate and engraft within the injured endothelium to promote vascular repair, possibly in combination with the secretion of trophic mediators.

### 2.2. ECFCs and Ischemic Diseases: An Alternative Strategy to Induce Therapeutic Angiogenesis

ECFCs might represent a potential cellular substrate to induce neovascularization in patients affected by various life-threatening cardiovascular disorders (Table 1) such as traumatic brain injury or stroke [61,62], retinal ischemia [21], AMI [63], and PAD [57,64]. In addition, preclinical investigations disclosed the therapeutic potential of ECFC transplantation to treat bronchopulmonary dysplasia [65] and ischemic kidney injury [66,67].

#### 2.2.1. Ischemic Brain Disease

Ischemic stroke is a common cause of morbidity and mortality worldwide, and it stems from the occlusion of an artery supplying oxygen and nutrients to the brain [62], which can undergo irreversible damage even after a short interruption of cerebral blood flow (CBF) [68]. Only a restricted (<10%) number of patients experiencing an ischemic stroke display a significant increase in CBF following thrombolytic treatment [69]. Therefore, there is the compelling need for alternative strategies to induce vascular repair and to restore blood flow more effectively in a larger cohort of patients. As recently reviewed in [62] and [70], ECFCs hold promise as a novel therapeutic option to promote vascular repair and effectively treat ischemic stroke. For instance, Ding and coworkers exploited bioluminescence imaging (BLI) to examine the reparative effect of ECFCs infected with a lentivirus carrying enhanced green fluorescent protein (eGFP) and firefly luciferase (Luc2) double fusion reporter gene in a murine model of ischemic stroke [61]. Then, labeled ECFCs were delivered via left cardiac ventricle injection in a mouse model of photothrombotic brain stroke [61]. Analysis of bioluminescence revealed that ECFCs homed to the brain on day 1 after the injection and persisted therein until days 4 and 7, although BLI signals progressively faded until disappearing at day 14. In agreement with these data, immunofluorescence staining revealed GFP-positive ECFCs at the infarct border, thereby supporting the evidence that ECFCs successfully homed to the ischemic brain [61]. ECFC transplantation proved to be an effective strategy to rescue brain damage and neurological disability by improving angiogenesis, decreasing neuronal apoptosis, and enhancing neurogenesis [61]. Likewise, the intrafemoral injection of oxine-labeled ECFCs in a rat model of brain ischemia, i.e., middle cerebral artery occlusion (MCAO), increased capillary density and favored neurogenesis within the ischemic site [71]. ECFCs did not integrate within neovasculature but stimulated neovessel growth by releasing soluble mediators, including VEGF, epidermal growth factor (EGF), and angiopoietin 2 [71].

In addition to stroke, ECFCs provided a suitable cellular substrate to promote vascular repair in a rodent model of traumatic brain injury (TBI) induced by lateral fluid percussion injury. ECFCs were intravenously infused in female rats 1 h after the injury, and their intravascular engraftment was traced using a male-specific Y chromosome probe in fluorescence in situ hybridization (FISH) [72]. ECFCs were detected within the inner cellular lining of the microvessels running through the injured area at 24 h, and their rate of engraftment further increased at 72 h after TBI. This resulted in an increase in capillary density and paracrine release of pro-angiogenic cues, such as VEGF and stromal-derived factor-1α (SDF-1α). Of note, ECFC transplantation caused a remarkable improvement of neurological disability starting at day 7 and persisting until day 14 after ECFC injection [72]. A follow-up investigation revealed that the intracerebroventricular injection of ECFCs in a mice model of TBI prevented disruption of the BBB and induced an increase in microvascular density, thereby rescuing neurological deficits [73]. Finally, ECFC transfusion in a rat model of a cerebral aneurysm (CA) dampened vascular remodeling and inflammation, which underlie the life-threatening consequences of CA rupture. ECFCs conferred protection against degeneration of the aneurysmal wall by preventing matrix metalloproteinase (MMP) activation, attenuating inflammatory signaling (e.g., nuclear factor κB (NF-κB), vascular cell adhesion molecule-1 (VCAM-1), inducible nitric oxide (NO) synthase (iNOS)), and protecting vascular smooth muscle cells (VSMCs) from apoptosis [54]. Altogether, these observations strongly suggest that ECFCs represent a promising cellular substrate to promote vascular repair in the brain (Table 1).

#### 2.2.2. Ischemic Retinopathy

The main leading cause of visual impairment is represented by ischemic retinopathies, such as diabetic retinopathy, retinal vein occlusion, and retinopathy of prematurity, which are all featured by vasodegeneration [74]. Preclinical evidence shows that ECFC transplantation could provide a promising approach to treat such disorders [21]. Unlike other organs, the retina benefits from the so-called immune privilege, which has evolved to mitigate local immune and inflammatory responses and preserve vision [75]. Therefore, the retina provides a suitable microenvironment to reliably evaluate the regenerative outcome of human ECFCs [70]. Intravital injection of ECFCs into a murine model of retinal ischemia promoted vascular regeneration by, respectively, decreasing the retinal avascular area (41%) and enhancing the normovascular area (30%). In addition, human ECFCs integrated within intraretinal microvasculature and engrafted within neovessels emerging within the ischemic areas [76]. Likewise, hiPSCs-derived ECFCs induced neovessel formation and decreased neovascular tufts in a similar murine model of retinal ischemia [42]. The regenerative outcome of ECFCs on the ischemic retina could be improved by enhancing pro-angiogenic signaling pathways. For instance, ECFCs-induced neovascularization in a murine model of diabetic retinopathy was enhanced by the previous intravitreal injection of the adeno-associated virus serotype 2 encoding a more stable, soluble, and potent form of angiopoietin 1 known as cartilage oligomeric matrix protein Ang1 (AAV2.COMP-Ang1) [77]. The reparative outcome of ECFC transplantation was also enhanced by alleviating the pro-inflammatory milieu typical of the ischemic retina. The systemic delivery of ARA290 (cibinetide), a peptide-based on the Helix-B domain of erythropoietin (EPO), exerted a local anti-inflammatory effect in a murine model of oxygen-induced retinopathy (OIR), thereby boosting ECFCs-induced neovascularization [78]. In addition to integrating within emerging neovessels, ECFCs can support retinal neovascularization in a paracrine manner. A recent investigation revealed that intravitreally injected ECFCs promoted vascular repair in a mouse model of OIR by releasing a wealth of angiocrine cues, including VEGF, interleukin-8, and insulin-like growth factor-binding proteins (IGFBPs). A careful evaluation of the most efficient ECFC phenotype disclosed an ECFC subset expressing high levels of CD44, which serves as a hyaluronic acid receptor. Notably, this vasoreparative effect was mimicked by administrating either conditioned media from ECFCs or human recombinant IGFBPs [79]. Furthermore, ECFCs stimulated retinal regeneration by supporting the vascular development promoted by bone marrow-derived CD34 hematopoietic stem cells, as recently demonstrated in a murine model of OIR [80]. A follow-up study revealed that the injection of ECFCs-derived extracellular vesicles restored the vascular network in a murine model of OIR by stimulating local angiogenesis through the delivery of a package of multiple microRNAs (miRNAs) [81]. Finally, a recent preclinical study evaluated some parameters that are crucial to support the clinical translation of ECFCs-based therapy in human patients. Preliminary evidence demonstrated the following in a murine model of OIR: (1) low (1 × 10^3^) ECFC doses caused a similar reduction in the avascular areas as compared to higher (1 × 10^4^, 1 × 10^5^) ECFC doses [21]; (2) systemic delivery of ECFCs through the common carotid artery was as effective as local intravitreal injection in inducing vasoreparation [21]; and (3) intravitreal injection of ECFCs into healthy eyes did not induce any inflammatory response and did not promote malignant growth [21]. Altogether, these findings endorse the emerging view that ECFCs provide a suitable cellular substrate for cell based-therapy of ischemic retinopathy (Table 1).

#### 2.2.3. Ischemic Cardiovascular Disorders

Ischemic heart disease is one of the most common types of CVD and represents the leading cause of mortality worldwide [2]. Therapeutic angiogenesis may provide a promising approach to promote myocardial revascularization and rescue cardiac contraction upon an ischemic insult [82,83]. Bone marrow-derived CD34^+^ hematopoietic stem cells, which include MACs, mitigated infarct size, increased coronary microvascular density, and rescued left ventricular ejection fraction (LVEF) in rodent models of AMI (Table 1) [84,85]. Therefore, a number of clinical trials were conducted to assess the impact of CD34^+^ cell transplantation in human patients [83]. This approach did not reveal any major hurdle and proved to be safe and feasible in several ischemic disorders, including CAD [86,87], AMI [85,88], refractory angina [89,90], and heart failure [91,92]. Nevertheless, careful evaluation of the therapeutic outcome of CD34^+^ cell infusion provided inconsistent results, as larger randomized controlled trials failed to detect a significant improvement in coronary regrowth and myocardial angiogenesis following AMI [82]. It has been postulated that the reparative phenotype of the cellular substrate to be transplanted in ischemic patients, along with other crucial issues (e.g., route of delivery, timing, and dose), is crucial for the successful outcome of therapeutic angiogenesis. ECFCs, which display the higher clonogenic potential among CD34^+^ cells, represent the most promising therapeutic agent to restore cardiac vasculature in ischemic diseases [10,93]. In addition, ECFCs may stimulate vascular regrowth by exploiting multiple cellular mechanisms, while MACs may only act as paracrine reservoirs [16]. An early study demonstrated that the infusion of autologous ECFC-like cells attenuated adverse myocardial remodeling in a porcine model of AMI by diminishing infarct size and increasing microvessel density. Interestingly, beta-galactosidase-labeled ECFCs physically engrafted within neovessels emerging at the border zone of the infarcted area, while they did not promote cardiomyogenesis [94]. In addition, it was proposed that ECFCs fostered cardiac neovascularization by releasing pro-angiogenic mediators, such as placental growth factor (PlGF) [94]. A subsequent study assessed the regenerative efficacy of ECFCs combined with mesenchymal progenitor cells (MPCs) in a rat model of AMI. Luciferase-labeled ECFCs were directly injected into the ischemic myocardium, and luciferase activity was evaluated until day 14. This strategy revealed that at least 1500 ECFCs were still retained in the beating heart at 2 weeks post-injection, albeit at a lower percentage as compared to day 1 [63]. Furthermore, ECFCs assembled into functional vessels that nicely integrated with the host vasculature and remained perfused at 3 months post-injection. Finally, ECFC transplantation mitigated left ventricular remodeling and dysfunction, thereby confirming the beneficial outcome of this approach on ischemia–reperfusion injury [63]. Recent evidence revealed that the injection of ECFCs into the peri-infarct region of an AMI model of severe combined immunodeficiency mice attenuated adverse post-AMI remodeling, presumably through paracrine effects [95]. The repair of cardiac injury was improved by the increase in capillary density and in the number of resident Sca1(+) cardiac progenitor cells [95]. Likewise, the direct injection of ECFCs and MSCs exerts a regenerative effect on the infarct area in a mouse model of AMI, which is seemingly mediated by the release of paracrine signals [96]. The feasibility of using autologous ECFCs to induce therapeutic angiogenesis in ischemic cardiac disorders is also suggested by several studies conducted on AMI patients. For instance, ECFC frequency in peripheral blood undergoes a rapid increase upon AMI, possibly in an attempt to repair the injured coronary microvasculature [97]. ECFC frequency in peripheral blood inversely correlated with microvascular obstruction at day 5 after AMI and positively correlated with the improvement in LVEF and the reduction in infarct size at 6 months [98]. It has been suggested that an increase in circulating levels of platelet-derived growth factor-BB (PDGF-BB could promote ECFC mobilization in AMI patients [99]. This evidence hints at the hyper-activation of endogenous ECFCs as an alternative strategy for cardiac repair to be pursued along with cell autologous cell transplantation.

#### 2.2.4. Peripheral Artery Disease and Critical Limb Ischemia

Peripheral artery disease (PAD) may present as intermittent claudication or critical limb ischemia (CLI) and may lead to amputation and poor life expectancy in patients who are not amenable for pharmacological treatment or surgical revascularization [100]. PAD is caused by the occlusion of the arterial conduits serving the lower extremities, which is due to the buildup of atherosclerotic plaques. CLI represents the end stage of PAD and is the manifestation of severe endothelial dysfunction upon the prolonged exposure to atherogenic risk factors, e.g., aging, diabetes, hypercholesterolemia, and smoking [100,101]. CLI may lead to the diminution or blockage of blood flow through the obstructed artery even at rest [100,101], thereby resulting in leg pain, non-healing ulcers, gangrene, or tissue loss [100]. The Inter-Society Consensus for the Management of PAD estimated that 25% of patients affected by CLI would die within 1 year from the diagnosis, whereas limb salvage will not be possible in an additional 30% of subjects [100]. Currently, available treatments for PAD patients are limited in the presence of severe cardiovascular co-morbidities, sepsis or gangrene, or in non-ambulatory individuals [102]. Consequently, there is a compelling need for new approaches to stimulate neovascularization through therapeutic angiogenesis. The potential efficacy of ECFC transplantation in PAD has been revealed by several preclinical investigations utilizing a murine model of CLI [70]. Primarily, the intravenous infusion of GFP-labeled ECFCs resulted in a significant rescue of blood perfusion at 14 days post-injection due to rapid (within 6 h) ECFC relocation to the ischemic hindlimb [93]. The regenerative efficacy of ECFCs was boosted when they were transplanted in combination with MSCs. It was further shown that ECFCs promoted neovascularization by physical incorporation within neovessels and through paracrine signaling [93]. Notably, ECFCs were transduced with a lentivirus expressing the thymidine kinase suicide gene driven by the endothelial-specific VEGF receptor 2 (VEGFR2) promoter; ganciclovir-mediated elimination of VEGFR2^+^ cells impaired neovascularization and blood flow recovery, thereby confirming the therapeutic efficacy of ECFC transplantation. Further evidence revealed that concomitant administration of MACs-secreted factors and ECFCs resulted in a three-fold enhancement of local cell retention in vivo and improved muscle perfusion, vessel maturation, and hindlimb regeneration [103]. Of note, the paracrine signals generated by MACs improved the regenerative outcome of ECFCs by stabilizing the neovessels [103]. The interaction among ECFCs and MSCs, which may differentiate into perivascular mural cells, was confirmed by a subsequent investigation, which demonstrated that ECFCs stimulate MSC incorporation via PDGF-BB/PDGF receptor-β signaling [54]. In addition, the genetic silencing of endoglin, an RGD-containing binding site for β1 integrins that regulates endothelial adhesion to mural cells [104], prevented the cooperation between co-delivered ECFCs and MSCs, thereby inhibiting neovascularization in a murine model of CLI [105]. Likewise, the intramuscular injection of ECFCs in combination with mesenchymal progenitor cells (MPCs) promoted neovascularization and blood flow recovery in ischemic skeletal muscles as compared to ECFCs or MPCs alone [57]. BLI further revealed that ECFCs + MPCs were still detectable in hind limbs at 14 days post-injection. Of note, the restoration of blood flow by ECFCs and MPCs required endogenous pro-angiogenic myeloid cells, such as Gr-1^+^ neutrophils and monocytes [57]. Finally, a recent investigation demonstrated that hiPSCs-derived ECFCs were able to promote neovessel formation and rescue blood perfusion in ischemic hind limbs [42]. This evidence hints at the possibility to exploit patient-derived hiPSCs to obtain a therapeutically relevant amount of ECFCs (>trillion in less than 3 months) to treat PAD and, possibly, other life-threatening ischemic disorders [42]. Furthermore, this investigation demonstrated that neuropilin-1 (NRP-1)-mediated activation of VEGFR2 signaling was able to enhance the proliferation rate and reduce senescence in late passage ECFC cultures (Table 1) [42].

## 3. Manipulation of Pro-Angiogenic Signaling Pathways to Improve ECFC Efficiency in Ischemic Diseases

ECFCs are regarded among the most suitable cellular substrates for therapeutic angiogenesis [15,16,50,59,70,110]. Yet, they are far from being probed in clinical trials on ischemic patients. A recent article surveyed the clinical studies that probed EPCs as a therapeutic agent against an array of pathologies [111]. Only 26 out of 341 clinical trials identified by searching the term “EPC” assessed the regenerative outcome of EPCs in severe ischemic disorders, such as CAD, PAD, and stroke, whereas the remaining investigations were of a more observational nature (i.e., they evaluated EPC levels as biomarkers for different pathological conditions). Five additional trials, which were identified through a search on PubMed and Web of Science databases, were conducted to examine the impact of EPCs-based therapy in PAD, CAD, pulmonary hypertension, and liver cirrhosis [111]. These studies demonstrated that EPC therapy was feasible and safe; however, it was concluded that the EPC populations employed, deriving either from bone marrow or UCB, were largely heterogenous and mainly consisted of hematopoietic angiogenic cells, such as MACs [33,111]. Surprisingly, no clinical trial has yet assessed the therapeutic efficacy and feasibility of ECFCs as cellular substrate to treat ischemic disorders. Several reasons have been proposed to explain why therapeutic angiogenesis via the transplantation of autologous ECFCs, which represent the only known truly endothelial precursor, is yet to be probed in ischemic patients.

Firstly, the frequency of circulating ECFCs is rather low, ranging from 0.28–15 ECFCs/10^7^ [112,113] to 1 ECFC/10^6^–10^8^ MNCs [59]. This amount of ECFCs is insufficient to achieve a therapeutically relevant number of cells for regenerative purposes [59,110,114,115].

Secondly, the in vivo mobilization of larger amounts of ECFCs in ischemic patients may not produce effective therapeutic outcomes [115]. Although an increase in circulating ECFCs has been observed upon an ischemic insult, such as AMI [97], CAD [116], CLI [117], forearm ischemia [118], and pulmonary hypertension [119], their angiogenic activity may be severely compromised [117,118,119,120].

Third, ECFCs’ angiogenic behavior is typically compromised in the presence of cardiovascular risk factors that predispose to the onset of ischemic disorders; this further contributes to discouraging the use of circulating ECFCs for autologous cell therapy [15,59,110,115]. For instance, the frequency and/or vasoreparative potential of circulating ECFCs are remarkably reduced with CAD [120], aging [121], diabetes [122], abdominal aortic aneurysm [123], venous thromboembolic disease [124], congenital diaphragmatic hernia [125], systemic lupus erythematosus [126], and the pediatric Moyamoya disease (MMD) [127]. In addition, ECFCs’ angiogenic behaviour is impaired in patients affected by human immunodeficiency virus (HIV), which has long been associated with cardiovascular disorders [128].

Fourth, ECFCs’ angiogenic activity could be further reduced by the harsh microenvironment of ischemic tissues. For instance, ECFC proliferation and tube formation are affected in the presence of elevated pro-inflammatory signaling [129], oxidative stress [130], hypoxia [131,132], and damage-associated molecular patterns (DAMP), such as extracellular histones [133] and monosodium urate [109].

Fifth, the therapeutic use of UCB-derived ECFCs, which display higher proliferative potential (up to 100 population doublings), enhanced angiogenic activity and greater telomerase activity as compared to circulating ECFCs [22,134], is currently not feasible [33]. On the one hand, the heterologous injection of umbilical cord blood-derived ECFCs may lead to an alloimmune response, both in vitro and in vivo, due to the expression of the of class I and class II major histocompatibility complex (MHC) molecules, which are, respectively, recognized by CD8^+^ and CD4^+^ T cells [135,136]. On the other hand, processing and freezing a sufficient amount of umbilical cord blood-derived ECFCs for each individual at birth, so that these could be banked until required during adulthood, could be exceedingly expensive at this moment [33].

In order to improve their therapeutic potential, several strategies may be adopted to expand large ECFC numbers in vitro and/or to cope with the harmful microenvironment of ischemic tissues and/or to manipulate specific pro-angiogenic signaling pathways to enhance their vasoreparative activity (Table 2, Table 3, Table 4, Table 5 and Table 6) [15,50,110].

### 3.1. Boosting ECFC Expansion Ex Vivo with Bioactive Cues

The paucity of ECFCs in peripheral blood represents a major limitation for their therapeutic use; nevertheless, several strategies have been designed to prime ECFCs ex vivo, thereby increasing their proliferation rate and achieving a therapeutically relevant cell dose (Table 2) [115]. For instance, long-term ECFC expansion can be accelerated when fetal bovine serum in the culture medium is replaced by human platelet lysate [137,138,139,140]. Consistently, a human platelet lysate gel has been recently designed to provide a 2D matrix enriched with growth factors and able to promote ECFC amplification and networking formation [141]. Likewise, ex vivo proliferation is enhanced by pre-conditioning ECFCs with the pro-angiogenic cues fucoidan [142,143,144] and SDF-1α [145,146], with the hexapeptide SFLLRN, which mimics the thrombin effect on PAR-1 [53,147,148], with the erythroid growth factor EPO [78,106,149,150,151,152], with the isoflavone genistein [153,154], with nicotine [155], and with an acidic growth medium [129,156]. Hypoxic pre-conditioning has also been proposed as an alternative approach to accelerate the ex vivo expansion of primary ECFCs [157,158], although subsequent studies demonstrated that their angiogenic activity is compromised under hypoxia [131,132]. Finally, ex vivo ECFC expansion could be achieved by stimulating an increase in intracellular Ca^2+^ concentration ([Ca^2+^]_i_), which has long been known to stimulate ECFC proliferation [159,160,161,162]. For instance, the liposomal delivery of nicotinic acid adenine dinucleotide phosphate (NAADP), the physiological agonist of endolysosomal two-pore channels (TPCs) [163,164,165,166], stimulated ECFC proliferation in a Ca^2+^-dependent manner [15]. Likewise, the lipid messenger arachidonic acid promoted ex vivo ECFC expansion by inducing extracellular Ca^2+^ entry through Transient Receptor Potential Vanilloid 4 (TRPV4) channel and then recruiting the endothelial NO synthase (eNOS) [167,168].

### 3.2. Priming Dysfunctional ECFCs to Rescue their Angiogenic Activity

Manipulation of the cell culture medium has also been exploited to rescue the angiogenic activity of dysfunctional ECFCs (Table 3). For instance, ECFCs derived from patients suffering from diabetes with neuroischemic (NI) or neuropathic foot ulcers display delayed colony emergence, reduced proliferative potential and migratory activity, impaired tubulogenic activity, and lower NO bioavailability [173]. Nevertheless, the glycomimetic C3 improved migration in both types of diabetic ECFCs and rescued bidimensional tube formation in NI ECFCs [173]. Of note, the glycomimetic C3 mimics the functions of heparan sulfate, a glycosaminoglycan that controls many pro-angiogenic signaling pathways and is remarkably shortened in ECFCs isolated from old individuals [174]. Thus, it could also be exploited to counteract the anti-angiogenic effect of aging on ECFCs. Parallel work conducted on an endothelial model of lipid-induced oxidative stress showed that the glycomimetic C3 was able to recruit the Nrf2/ARE and Akt/eNOS signaling pathways, thereby exerting a cytoprotective effect and rescuing NO release [175]. Similarly, pre-treatment of diabetic ECFCs with adiponectin rescued in vitro migration and neovessel formation in a mouse model of hindlimb ischemia under both normo- and hyperglycemic conditions [176]. Furthermore, pre-treatment with the low molecular weight fucoidan, which is extracted from the brown algae *Ascophyllum nodosum*, counteracted ECFC senescence by rescuing in vitro proliferation and neovascularization in a mouse model of hindlimb ischemia [143]. Fucoidan exerted a protective effect by engaging the focal adhesion kinase (FAK)/Akt and FAK/extracellular-signal regulated kinase (ERK) pathways in senescent ECFCs [143]. In addition to chemical manipulation of the cell culture conditions, dysfunctional ECFCs could be primed by physical stimulation. For instance, a recent investigation revealed that far-infrared radiation treatment rescued motility and tube formation in ECFCs isolated from type 2 diabetes mellitus patients by reducing the microRNA-134 (miR-134-5p) and, consequently, up-regulating the nuclear receptor-interacting protein 1 (NRIP1) [177]. The same results were obtained on ECFCs exposed to high glucose (HG-ECFCs) [177]. NR1P1 was regulated by miR-134-5p at the transcriptional level and was found to regulate ECFC proliferation and tube formation [177]. In agreement with these observations, FIR pre-treatment enhanced HG-ECFCs-dependent neovascularization in a mouse model of hindlimb ischemia [177].

Ex vivo expanded ECFCs could also be exposed to a culture medium mimicking the hostile microenvironment of ischemic tissues. This approach has long been used to search for suitable bioactive compounds able to counteract ECFC dysfunction. For instance, ceramide 1-phosphate and acidic pre-conditioning prevented tumour necrosis factor α (TNFα) and monosodium urate crystal-induced ECFC death [109,129]. Of note, acidic pre-conditioning rescued ECFC proliferation, migration, and tube formation also in the presence of high inflammatory (TNFα) and high-glucose conditions [129]. In agreement with these observations, pre-conditioned ECFCs promoted an increase in capillary density, favored muscle regeneration, and rescued local blood flow in ischemic hindlimbs of normoglycemic and type 2 diabetic mice [129]. Similarly, the cholesterol-lowering drug, atorvastatin, rescued ECFC proliferation and tube formation in the presence of a highly oxidant microenvironment by up-regulating annexin A2, which is a Ca^2+^-regulated membrane binding protein that is involved in new blood vessel formation [178]. A recent investigation revealed that the hypoxia-induced impairment of angiogenic activity can be counteracted by pre-treating ECFCs with iptakalim [179]. Iptakalim is a lipophilic para-amino drug with a low-molecular weight that may serve as agonist of ATP-dependent K^+^ channels [180]. Iptakalim rescued ECFC proliferation and bidimensional tube formation in the presence of hypoxia by activating Akt and eNOS in a Ca^2+^-dependent manner [179,181]. In agreement with these observations, ATP-dependent K^+^ channels may promote endothelial hyperpolarization [182], thereby increasing the driving force that sustains extracellular Ca^2+^ entry in ECFCs [183]. The following increase in [Ca^2+^]_i_ may engage the Ca^2+^/Calmodulin-dependent protein kinase II (CaMKII) and induce CaMKII-dependent Akt phosphorylation [181]. In addition, extracellular Ca^2+^ entry drives the nuclear translocation of the Ca^2+^-sensitive transcription factor NF-κB, which stimulates ECFC proliferation and tube formation (Table 3) [184].

### 3.3. Priming Healthy ECFCs to Enhance their Angiogenic Activity

Pre-conditioning with bioactive cues has also been shown to boost ECFCs’ angiogenic activity. These maneuvers aim at favoring ECFC retention and survival within the noxious microenvironment of ischemic tissues and/or at stimulating ECFC proliferation and neovessel formation ability both in vitro and in mouse models of hindlimb ischemia [50,110]. Due to the limited access to dysfunctional ECFCs, most of the studies aiming at manipulating the pro-angiogenic signaling machinery for therapeutic purposes were conducted on umbilical cord blood-derived ECFCs or circulating ECFCs isolated from healthy donors (Table 4).

Ex vivo priming with SDF-1α stimulated ECFC adhesion to endothelial monolayers and tube formation in vitro as well as microvessel formation and blood flow increase in a murine model of hindlimb ischemia [145]. SDF-1α promoted matrix metalloprotease-2 (MMP-2) and fibroblast growth factor-2 (FGF-2) secretion from ECFCs via the direct interaction with the Gi-protein coupled receptor, CXCR4, and with cell surface heparan sulfate proteoglycans (HSPGs) [145]. Similarly, SDF-1α stimulated E-selectin mediated ECFC adhesion and migration following endotoxic endothelial injury in a CXCR4-dependent manner [146]. Additionally, SDF-1α promoted the Ca^2+^-dependent recruitment of the phosphatidylinositol 3-kinase (PI3K)/Akt and ERK signaling cascades, thereby further contributing to induce ECFC migration in vitro and neovessel formation in vivo [186].

Likewise, an earlier report showed that fucoidan promoted ECFC proliferation, migration, and tube formation in vitro; this latter effect was favored by co-stimulation with FGF-2 and was due to the up-regulation of the α6 integrin subunit of the laminin receptor [142]. A subsequent investigation revealed that fucoidan-primed ECFCs resulted in enhanced vascularization and blood flow recovery in a mouse model of hindlimb ischemia [144]. More recent work disclosed that fucoidan recruits the PI3K/Akt signaling pathway, thereby triggering a transcriptional program that leads to the expression of multiple genes involved in ECFC proliferation, migration, and cytoskeletal reorganization [169].

Ex vivo priming with EPO stimulated ECFC proliferation, migration, and tube formation in vitro, prevented H_2_O_2_-induced apoptosis, and favored ECFC survival in Matrigel plugs; furthermore, the transplantation of EPO-primed ECFCs caused a remarkable increase in capillary density of blood flow recovery in a murine model of hindlimb ischemia [149]. The pro-angiogenic effect of EPO was mediated by the EPO receptor (EPOR) and the CD131 receptor subunit [149]. Notably, the pre-conditioning effect of EPO was mimicked by ARA290, which is a selective agonist of the EPOR/CD131 complex [152]. A recent report showed that EPO primed ECFCs by recruiting the AMP-activated protein kinase (AMPK) and the downstream target Krüppel-like factor 2 (KLF2) [150]. In turn, KLF2 is a transcription factor that mediates ECFC differentiation and supports ECFCs-mediated neovascularization [187]. Furthermore, EPO has been shown to stimulate neovessel growth in murine models of hindlimb ischemia by promoting extracellular Ca^2+^ influx through Transient Receptor Potential Vanilloid 1 (TRPV1) [188]. ECFC priming with EPO significantly improved their regenerative outcome following MCAO. For instance, the systemic delivery of ECFCs primed with EPO was the most effective at restoring neurological function as compared to control ECFCs or EPO alone. EPO-primed ECFCs reduced infarct volume and apoptosis by stimulating angiogenesis and neurogenesis [106,151]. This observation was supported by an independent study, showing that EPO-primed ECFCs administered through the intratail vein efficiently homed to the ischemic hemisphere, thereby attenuating disruption of the blood–brain barrier (BBB), promoting neuronal survival, and enhancing cerebral blood flow (Table 4) [106].

Low doses (100 pM) of genistein, which is the most abundant isoflavone in soybeans, also induced ECFC proliferation and migration in vitro by activating, respectively, the ERK1/2 and the integrin-linked kinase (ILK) pathways [153]. The transplantation of genistein-primed ECFCs in a murine model of AMI enhanced neovascularization through an increase in ECFC proliferation and in the secretion of pro-angiogenic cytokines, such as SDF-1α and FGF-2 [153]. In turn, this mitigated adverse cardiac remodeling and improved left ventricular function, as measured by echocardiography [153]. Similarly, bone morphogenetic protein 4 (BMP4) stimulated ECFC proliferation and migration in vitro by promoting the up-regulation of VEGF, SDF-1α, and angiopoietin 2; thereafter, BMP4-primed ECFCs promoted neovascularization both in subcutaneously implanted Matrigel plugs and in a mouse model of hindlimb ischemia [189]. Acidic pre-conditioning (pH = 6.6 for 6 h) represents another effective strategy to prime ECFCs for therapeutic purposes. This maneuver stimulated ECFC proliferation, tube formation, and SDF-1α-induced chemotaxis by recruiting the PI3K/Akt and ERK1/2 signaling pathways [156]. In addition, it stimulated ECFCs to release a wealth of pro-angiogenic cues, such as VEGF, platelet-derived growth factor (PDGF), and basic fibroblast growth factor (bFGF) [129]. Furthermore, acidic pre-conditioning boosted limb reperfusion upon the systemic delivery of ECFCs [156].

Finally, hypoxic conditioning of the human amniotic fluid-derived stem cells conditioned medium was recently exploited to promote revascularization and rescue LVEF in a murine model of AMI [190]. In vitro analysis revealed that the human amniotic fluid stem cell secretome stimulated human ECFCs to undergo bidimensional tube formation through an oscillatory increase in [Ca^2+^]_i_ [190,191], which represents the main signaling mode whereby VEGF stimulates ECFCs to undergo tubulogenesis (Table 4) [160,161,184].

An additional array of ex vivo priming strategies was found to stimulate ECFCs’ angiogenic activity in vitro, but their outcome was not probed in vivo. For instance, the glycoprotein thrombospondin 1 (TSP1) is an extracellular matrix glycoprotein whose levels increase in the plasma of PAD patients and in ischemic limbs [192]. Recombinant human TSP-1 reduced ECFC proliferation and tube formation and impaired their pro-angiogenic potential in a mouse model of hindlimb ischemia [192]. Nonetheless, ex vivo priming of ECFCs with the TSP-Hep I peptide, which mimics the NH_2_-terminal tail of TSP-1, increased ECFC adhesion to an endothelial monolayer, which is the prerequisite for physical engraftment within neovessels, by up-regulating syndecan 4 and α6-integrin [193]. Furthermore, the TSP-Hep I peptide stimulated ECFC proliferation, tube formation, and bidimensional tube formation, and these observations further endorse the view that it could be efficiently used to prime ECFCs before autologous transplantation [193]. Likewise, recombinant osteoprotegerin (OPG), which functions as trap receptor by binding receptor activator nuclear factor Κb ligand (RANKL), stimulated ECFC tube formation in vitro by promoting SDF-1α expression and by recruiting the ERK and PI3K/Akt/mammalian target of rapamycin (mTOR) signaling pathways [194]. A subsequent investigation confirmed that OPG induced the expression of pro-angiogenic genes, including hypoxia-up-regulated protein, catenin alpha-1, and protein disulfide-isomerase A3, and it reduced the expression of pro-apoptotic genes, such as Taldo1 and Sh3glb1 [195]. The in vivo Matrigel plug assay confirmed that locally injected OPG promoted neovessel formation by endogenous vascular endothelial cells and EPCs [194,195]. Nevertheless, it was not assessed whether this procedure was also able to boost the engraftment of systemically delivered ECFCs.

Finally, recent investigations revealed that the soluble form of CD146 (sCD146), a component of the endothelial junction machinery that controls paracellular permeability and monocyte transmigration, could be used to effectively prime ECFCs before transplantation. Single photon emission computed tomography-CT (SPECT-CT) imaging revealed that ex vivo priming with sCD146 did not increase ECFC homing in a murine model of hindlimb ischemia, but it improved ECFC survival and increased neovascularization [196]. In vitro analysis revealed that sCD146 acts through a membrane signalosome located within lipid rafts and containing the short isoform of CD146 (shCD146), angiomotin, presenilin-1 (PS-1), VEGFR1, and VEGFR2. sCD146 binds to angiomotin, thereby recruiting MMPs and a disintegrin and metalloproteinase (ADAM) to cleave the extracellular part of shCD146, i.e., sCD146, which is released in the extracellular medium and amplifies sCD146 signaling. Thereafter, PS-1 cleaves the intracellular fragment of shCD146 (shCD146 IT), which translocates into the nucleus where it associates with the transcription factor CSL and regulates the expression of genes underlying endothelial cell survival (i.e., FADD and Bcl-xl) and angiogenesis (i.e., eNOS and shCD146) [196]. shCD146-induced transcription of eNOS and Bcl-xl required both VEGFR1 and VEGFR2 [196], which is a finding that is consistent with the notion that CD146 may serve as VEGFR2 co-receptor [197]. A subsequent investigation confirmed that pre-treating ECFCs with sCD146 enhanced in vitro proliferation and clonogenic by shortening the onset time of the colonies [197]. Moreover, this treatment interfered with cellular senescence, maintained a stem cell phenotype, and partially favored the endothelial–mesenchymal transition by inducing the expression of embryonic transcription factors. sCD146 exerted these pro-angiogenic effects by up-regulating miR-21. In agreement with these observations, the in vivo injection of sCD146-primed ECFCs isolated from PAD patients favored neovascularization and restored local blood flow in a mouse model of hindlimb ischaemia [198]. Overall, these findings suggest that priming ECFCs with sCD146 could constitute a promising strategy to improve the use of autologous ECFCs to treat ischemic disease (Table 4).

### 3.4. Local Injection of Chemoattractants Stimulates ECFC Homing to Sites of Neovessel Formation

The local injection of pro-angiogenic cues has been widely exploited to promote ECFC homing and engraftment within ischemic regions. For instance, intramuscular injection of the formyl peptide receptor-2 (FPR2) agonist WKYMVm combined with the intravenous delivery of ECFCs enhanced neovascularization and blood perfusion, reduced muscle necrosis, and improved ischemic limb salvage in a mouse model of hindlimb ischemia [199]. WKYMVm acted through FPR2 to stimulate ECFC proliferation, migration, and tube formation [199]. Likewise, the intramuscular injection of recombinant periostin, an extracellular matrix protein, favored ECFC recruitment, blood flow perfusion, and leg salvage in a murine model of hindlimb ischemia [200]. Periostin was able to stimulate ECFC proliferation, adhesion, and tube formation through the first fasciclin-1 (FAS-1) domain, which signaled through β3 and β5 integrins [200]. A follow-up study revealed that a synthetic peptide, encompassing 136–156 amino acids of the FAS-1 domain, stimulated ECFC proliferation, migration, and tube formation by recruiting the ERK and PI3K/Akt signaling pathways [201]. Thus, the local injection of this periostin peptide could enhance the regenerative outcome of ECFC transplantation. A recent investigation showed that local injection of ceramide 1-phosphate enhanced ECFCs-dependent neovascularization and muscle regeneration in a murine model of hindlimb ischemia [109]. Ceramide 1-phosphate was also able to promote ECFC proliferation, migration, adhesion, and tube formation by recruiting the ERK1/2 and PI3K/Akt signaling pathways [109]. Finally, the local injection of sCD146 promoted ECFC homing in in vivo Matrigel plugs and in a rat model of hindlimb ischemia [202]. In vitro analysis confirmed that sCD146 stimulated ECFC migration and tube formation by up-regulating the pro-angiogenic genes eNOS, VEGFR2, MMP-2, and urokinase-type plasminogen activator (uPA) [202]. It is further likely that sCD146 promotes revascularization through the intracellular shedding of the shCD146, as described above [196].

### 3.5. Epigenetic Reprogramming of ECFCs

Current evidence supports an association between epigenetic defects and CVD. Epigenetic alterations consist in changes in gene expression that do not involve modifications (mutations) of the primary DNA sequence [203], these include DNA methylation [204]; post-translational histone modifications, such as acetylation and methylation [205], and non-coding RNA or microRNA-mediated post-transcriptional regulation of gene expression [206].

It has been postulated that epigenetic drugs might derepress crucial proangiogenic signaling pathways, thereby improving ECFC’s vasculogenic ability in vivo (Table 6) [207,208,209]. For instance, ex vivo pre-treatment with trichostatin, a histone deacetylase (HDAC) inhibitor, enhanced ECFCs-mediated revascularization and blood flow recovery in a murine model of hindlimb ischemia [210]. Trichostatin acted by up-regulating the transcription factor TAL1/SCL, which, in turn, recruited the histone acetyltransferase p300 to induce the expression of an array of pro-angiogenic genes involved in endothelial motility and adhesion [210]. Likewise, pre-treating ECFCs with a cocktail of epigenetic drugs, such as GSK-343, which inhibits methyltransferase EZH2 to relieve H3K27me2/3-mediated gene silencing, and the HDAC inhibitor panobinostat, improved ECFC migration, bidimensional tube formation, and resistance to serum starvation-induced apoptosis [207]. Furthermore, RNA-seq combined with gene ontology (GO) analysis revealed that this combination of epigenetic drugs induced a gene transcriptional program driving endothelial cell migration, adhesion, and neovessel formation [207]. Of note, the pharmacological blockade of EZH2 and HDAC improved ECFCs-derived neovessel formation in vivo and rescued local blood flow in a murine model of hindlimb ischemia [207].

Epigenetic drugs were also shown to rescue pro-angiogenic activity in dysfunctional ECFCs. For instance, panobinostat treatment was more effective in restoring capillary-like network formation both in vitro and in vivo in ECFCs obtained by children affected by MMD as compared to control cells [211]. Panobinostat acted by derepressing retinaldehyde dehydrogenase 2 (RALDH2) expression, which is severely compromised in MMD [212].

In parallel with histone modifications, miRNAs are involved in the epigenetic regulation of ECFCs and could contribute to repress their pro-angiogenic activity in the presence of cardiovascular risk factors. Thus, the pharmacological inhibition of miRNAs-dependent epigenetic silencing represents an alternative strategy to improve ECFC’s therapeutic potential. For instance, a recent investigation disclosed that miR-146a-5p and miR-146b-5p were up-regulated in ECFCs isolated from CAD patients, thereby dampening ECFC migration and tube formation [213]. Likewise, far-infrared irradiation (FIR) treatment rescued in vitro migration and tube formation by down-regulating miR-486-5p expression in CAD ECFCs (Table 6) [185].

## 4. Genetic Manipulation of Pro-Angiogenic Signaling Pathways in ECFCs

An additional strategy to rescue the vasoreparative phenotype of dysfunctional ECFCs consists in the genetic manipulation of their pro-angiogenic signaling machinery [15,50,110]. It has been widely demonstrated that enhancing ECFC ability to perceive and decode pro-angiogenic cues significantly improves their regenerative outcome in ischemic disorders. These evidence are consistent with the notion that ECFCs display high genomic stability in cell culture and are more amenable than MACs to genetic manipulation [15,50,214]. Earlier reports showed that transducing MACs with an adenovirus encoding for VEGF gene reduced by ≈30 times the dose required to restore blood flow [13,215]. Therefore, it is conceivable that the genetic manipulation of ECFCs could lead to even more effective therapeutic outcomes. Preliminary studies revealed that ECFCs can be engineered to deliver recombinant proteins to treat anemia [216], hemophilia A [217,218], or von Willebrand disease [219,220] or to secrete angiogenic inhibitors [221,222]. When ECFCs were engineered with non-viral plasmids encoding for human coagulation factor VIII (FVIII), they mainly engrafted within the bone marrow and spleen of NOD/SCID (non-obese diabetes/severe combined immunodeficiency) mice and retained both transgene expression and the endothelial phenotype; in addition, they persisted in secreting in circulation therapeutic levels of FVIII even at 5 months after transplantation [217]. Nevertheless, plasmid vectors display a lower efficiency of transfection as compared to viral vectors, such as lentiviruses, adenoviruses, and adeno-associated viruses [223]. In addition, plasmids constitute non-replicating episomes, so that transgene expression is transient and the payload is progressively diluted and ultimately lost by cell division [224]. However, it has been widely demonstrated that ECFCs may be easily transfected with viral vectors, including lentiviruses [50,214,225], which integrate into the host genome of both dividing and non-dividing cells, thereby resulting in stable transgene expression [226]. For instance, canine ECFCs engineered with a lentiviral vector encoding for the canine FVIII (cFVIII) transgene and then subcutaneously implanted in Matrigel plugs induced the therapeutic expression of cFVIII for 27 weeks [227]. Subsequently, cFVIII-overexpressing autologous ECFCs were implanted in the omentum of a canine model of hemophilia A, thereby secreting into circulation therapeutic levels of cFVIII up to 250 days after the transplantation [218,228]. Similarly, Melero-Martin’s group transfected umbilical cord blood-derived ECFCs with a tetracycline-regulated lentivirus encoding for the EPO transgene and subcutaneously implanted bioengineered ECFCs with MSCs into nude mice [216]. Genetically modified ECFCs gave rise to the de novo formation of blood vessels and actively stimulated erythropoiesis in immunodeficient animals [216]. Furthermore, ECFCs-mediated gene delivery rescued erythropoietin levels, also in the bloodstream of anemic mice [216]. These and other [219] studies suggested that the genetic manipulation of ex vivo ECFCs could remarkably improve their regenerative potential. Herein, we survey how the genetic manipulation of specific pro-angiogenic and/or pro-survival signaling pathways enhances ECFCs-mediated neovascularization (Table 6).

An earlier report showed that the angiogenic activity of ECFCs harvested from South Asian (SA) men was compromised as compared to their matched European controls: colony-forming ability, in vitro migration, and bidimensional tube formation, as well as Akt1 and eNOS activity, were impaired in SA men-derived ECFCs [229]. As a consequence, ECFC transplantation in immunodeficient mice failed to promote re-endothelialization in wire-injured femoral arteries and after hindlimb ischemia [229]. However, lentiviral vector gene delivery of a constitutively active Akt1 mutant was able to rescue the pro-angiogenic activity of SA men-derived ECFCs by recruiting eNOS, thereby restoring their therapeutic outcome in vivo [229]. A slightly different approach consisted in forcing ECFCs to overexpress integrin β1 [230], which promotes ECFC binding to extracellular matrix proteins in ischemic tissues [231] and further exerts independent angiogenic effects [232]. ECFCs were engineered with a lentivirus encoding for murine integrin β1 and injected, either locally or intravenously, in a murine model of hindlimb ischemia [230]. The transplantation of integrin β1-overexpressing ECFCs restored the local vascular network by physically engrafting within neovessels and rescued local blood flow [230]. A recent investigation focused on NADPH oxidase 4 (NOX4) [233], which is constitutively active and promotes H_2_O_2_ production, thereby stimulating ECFC proliferation and migration [234]. NOX4 overexpression stimulated ECFC migration in vitro, albeit this effect was mediated by enhanced anion superoxide, rather than H_2_O_2_, production. In agreement with this observation, local injections of these genetically manipulated ECFCs enhanced tissue revascularization and local blood flow in a murine model of hindlimb ischemia [233]. In addition, NOX4 overexpression in ECFCs promoted neovascularization by driving the expression of multiple pro-angiogenic genes, such as eNOS, SDF-1α, MMP-9, and dipeptidyl peptidase IV [233]. A recent investigation revealed that adenoviral overexpression of the transcription factor Forkhead Box F1 (FOXF1), which is up-regulated in ECFCs as compared to vascular endothelial cells, induced basal sprouting even without VEGF stimulation [235]. FOXF1 overexpression induced the up-regulation of the Notch2 receptor, which plays a crucial role in controlling tip cell generation. Furthermore, the genetic manipulation of FOXF1 levels increased the expression of VEGFR2 as well as of the arterial endothelial cell marker ephrin B2, while it down-regulated the venous marker EphB4 [235]. These findings support the notion that circulating ECFCs engraft within neovessels and thereafter initiate sprouts by forming tip cells. A series of investigations suggested that genetic manipulation of the Ca^2+^ handling machinery might represent an alternative strategy to boost ECFC’s regenerative potential (Table 6) [15,134,159]. It has been suggested that the stronger pro-angiogenic response induced by VEGF in umbilical cord blood-derived ECFCs as compared to their circulating counterparts is driven by the selective expression of the plasmalemmal channel, Transient Receptor Potential Channel 3 (TRPC3) [9,236]. Thus, transducing autologous ECFCs with TRPC3 could rejuvenate their reparative potential and improve the outcome of cell-based therapy [15,134,237].

An alternative approach consists of repressing anti-angiogenic pathways. For instance, the genetic inhibition of Egfl7 (VE-statin), a 30 kDa protein that regulates endothelial cell fate and angiogenesis, improved ECFC proliferation, migration, and tube formation in vitro and enhanced blood flow recovery in a murine model of hindlimb ischemia in vivo [238]. It has been documented that the inhibition of protease-activated receptor 1 (PAR1) and PAR2 increases VEGF-induced ECFC’s proangiogenic activity [239]. Thus, the genetic suppression of PAR1 or PAR2 could improve ECFCs-dependent tubulogenesis, but this hypothesis remains to be experimentally probed.

## 5. Priming ECFCs with Mesenchymal Stem Cells

We have hitherto described an array of pharmacological and genetic strategies to manipulate crucial pro-angiogenic signaling pathways and improve ECFCs-mediated neovascularization. An alternative approach consists of inoculating ECFCs in combination with stem/progenitor cells, such as MSCs (Table 7) [110,240] and adipose-derived stem cells [58], or with perimural cells, such as vascular smooth muscle cells [241]. The mechanisms whereby co-injection with MSCs from multiple sources boosts ECFC’s regenerative potential have been widely investigated [70,110].

An earlier report revealed that MSCs isolated from four different murine tissues, i.e., bone marrow, myocardium, skeletal muscle, and white adipose tissue, stimulate ECFC proliferation, migration, and tubulogenesis by secreting a plethora of pro-angiogenic cues [55]. These included VEGF, hepatocyte growth factor (HGF), and MMP-9. The subcutaneous implantation of ECFCs in combination with GFP-expressing MSCs remarkably increased neovessel density in immunodeficient mice; careful histological inspection revealed that ECFCs mainly lined the lumen of neovessels, whereas MSCs adopted a perimural position [55]. A recent study confirmed that bone marrow-derived MSCs facilitated ECFC engraftment within Matrigel plugs [242]. MSCs-primed ECFCs displayed reduced self-renewal potential but showed enhanced tubulogenic activity in vitro and neovessel formation capacity in vivo. MSCs modulated ECFC’s angiogenic activity via direct contact rather than in a paracrine manner; short-term contact (4 days) with MSCs activated the Notch signaling pathway, thereby priming ECFCs toward a mesenchymal phenotype that showed enhanced adhesive and tubulogenic properties [242]. Microarray gene expression analysis confirmed that ECFCs cultured in a MSC niche underwent the up-regulation of an array of genes encoding for cell cycle, adhesion, and extracellular matrix proteins [242]. Furthermore, bone marrow-derived MSCs stimulated ECFCs to proliferate, to form capillary-like structures in vitro, and to increase capillary density in the ischemic gastrocnemius muscle by up-regulating sphingosine kinase 1 (SphK1) expression and activity [243]. In turn, SphK1 promoted the endogenous production of sphingosine 1-phosphate (S1P), which autocrinally acted on S1P1 and S1P3 receptors to induce ECFC’s tubulogenic activity [243]. Moreover, bone marrow-derived MSCs improved ECFC proliferation, colony formation, migration, and angiogenic activity by secreting SDF-1α [244]. An independent investigation showed that MSCs did not need to release paracrine signals to promote a microenvironment conducive for ECFC network formation [245]. Conversely, adipose-derived mesenchymal stem cells primed ECFCs to undergo bidimensional tube formation in vitro and assembly into patent vessels in vivo, thereby restoring local blood in a murine model of hindlimb ischemia, by releasing multiple growth factors, including VEGF-A, bFGF, and PDGF-BB [246]. Therefore, priming ECFCs with MSCs represents a promising strategy to improve the therapeutic efficacy of autologous ECFCs in ischemic disorders (Table 7).

## 6. Conclusions

ECFCs represent the most suitable cellular substrate for the autologous therapy of ischemic disorders, but they have not yet been probed in clinical trials in human subjects. Translating the vasoreparative potential of circulating UCB-derived ECFCs to the patient’s bed requires unraveling the intracellular signaling pathways that drive their angiogenic activity. Herein, we surveyed the variety of strategies that are under investigation to improve ECFCs’ regenerative potential in patients affected by cardiovascular disorders or exposed to cardiovascular risk factors. This approach is imposed by the necessity to rescue their reduced capacity to form colonies, proliferate, migrate, engraft within neovessels, and survive in the harsh microenvironment of hypoxic tissues. Currently, ex vivo expanded ECFCs undergo either genetic or pharmacological manipulation to engage crucial pro-angiogenic signaling pathways, e.g., PI3K/Akt, ERK, eNOS, and Ca^2+^ signaling, to improve their ability to induce neovascularization in injured tissues. However, the reinoculation of autologous ECFCs requires the development of protocols compatible with the good manufacturing practice (GMP) for cell-based therapies and the use of xeno-free culture media, e.g., platelet lysate (Section 3.1), to achieve their expansion [50,70]. In our opinion, future work will have to explore the possibility to target endogenous ECFCs to favor their clonal expansion and rescue their regenerative potential directly in the patient. This could be achieved through the local delivery of genes encoding for growth factors, such as VEGF or FGF-4 [5], or for signaling components of their pro-angiogenic machinery, such as TRPC3 [134] or store-operated Ca^2+^ channels [114]. Interestingly, local injection of the human amniotic fluid stem cell secretome stimulated ECFCs-dependent cardiac revascularization in a murine model of AMI [190]. The delivery of cell-free products, such as extracellular vesicles endowed with a remarkable pro-angiogenic activity, could represent a therapeutically feasible strategy to engage circulating ECFCs and further promote their mobilization in ischemic patients. Finally, it has been proposed to stimulate therapeutic angiogenesis by implanting ex vivo expanded ECFCs in vascular network composed of biocompatible materials [50,110]. An alternative approach would be to exploit bioartificial structures that are able to recruit specific pro-angiogenic signaling pathways. For instance, a recent investigation revealed that optical stimulation with visible light (525 nm) of ECFCs plated on the photosensitive conjugated polymer regioregular poly (3-hexyl-thiophene) (P3HT) induced proliferation and tube formation by activating the non-selective Ca^2+^-permeable channel TRPV1 [248]. This strategy permits stimulating circulating ECFCs with no need of viral manipulation (as required by conventional optogenetics) and with unprecedented spatial and temporal resolution. Thus, it has been proposed that the injection of infrared-sensitive biocompatible and photosensitive nanoparticles combined with the local implantation of a source of visible light could represent a promising solution to treat ischemic disorders by recruiting resident ECFCs [188].

## Figures and Tables

**Figure 1 ijms-21-07406-f001:**
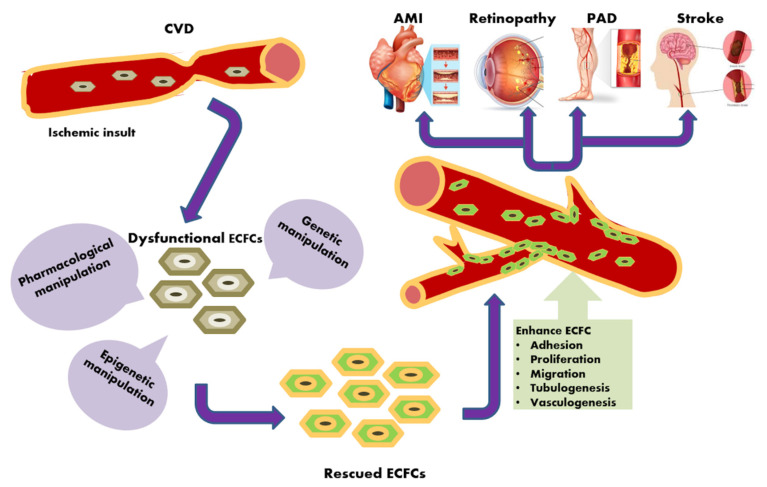
Manipulating dysfunctional endothelial colony-forming cells (ECFCs) to improve their regenerative potential for therapeutic angiogenesis. ECFCs isolated from the peripheral blood of individuals suffering from cardiovascular disease (CVD) present a reduced therapeutic efficacy. A number of strategies were designed to improve the regenerative potential of these ECFCs in the view of autologous cell-based therapy. These treatments include pharmacological pre-conditioning (e.g., with bioactive cues), genetic manipulation, and epigenetic activation, to improve their pro-angiogenic potential. It has been shown that ECFC manipulation remarkably improves neovascularizaiton and restores local blood flow in animal models of acute myocardial infarction (AMI), ischemic retinopathy, peripheral artery disease (PAD), and stroke.

**Table 1 ijms-21-07406-t001:** Preclinical applications of endothelial colony-forming cells (ECFCs).

Disease	ECFC Source	Effect	Reference
Ischemic stroke	UCB	Neurological functional recovery, improved angiogenesis and neurogenesis, and decreased apoptosis	[61]
Ischemic stroke	UCB	Rescue of BBB integrity, reduced cerebral apoptosis, increased CBF	[106]
Ischemic stroke	PB	Rescue of BBB structure and function	[107]
MCAO	UCB	Neurological functional recovery, improved angiogenesis and neurogenesis, reduced astrogliosis	[71]
TBI	UCB	Neurological functional recovery and improved angiogenesis	[72]
TBI	UCB	Rescue of BBB integrity and angiogenesis	[73,108]
CA	UCB	Protection against degeneration of the aneurysmal wall	[54]
OIR	PB	Vascular repair, decreased avascular areas, and increased normovascular areas	[76]
OIR	hiPSCs-derived ECFCs	Increased neovessel formation and decreased neovascular tufts	[42]
OIR	UCB	Improved angiogenesis and reduced avascular area	[81]
OIR	UCB	Vascular repair	[21,79]
OIR	UCB	Improved angiogenesis and vascular repair	[80]
Diabetic retinopathy	UCB	Stabilized vision and improved angiogenesis	[77]
AMI	UCB	Improved cardiac function, enhanced neovascularization, and decreased myocardial fibrosis	[10,93]
AMI	PB	Improved cardiac function and enhanced neovascularization	[95]
AMI	UCB	Improved cardiac function and enhanced neovascularization and reduced fibrosis	[96]
AMI	UCB	Improved cardiac function and mitigation of adverse remodeling	[63]
CLI	UCB	Blood flow restoration	[57]
CLI	UCB	Rescued of blood perfusion and enhanced neovascularization	[70,93]
CLI	hiPSCs-derived ECFCs	Promote neovessel formation and rescue blood perfusion	[42]
CLI	UCB	Improved muscle perfusion and hindlimb regeneration	[103]
CLI	UCB	Rescued of blood perfusion and enhanced neovascularization	[105]
CLI	UCB	Enhanced vascularization, leg reperfusion and muscle repair	[109]

AMI: acute myocardial infarction; BBB: blood–brain barrier; CA: cerebral aneurysm; CLI: critical limb ischemia; hiPSC: human pluripotent stem cells; MCAO: middle cerebral artery occlusion; OIR: oxygen-induced retinopathy; PB: peripheral blood; TBI: traumatic brain injury; UCB: umbilical cord blood.

**Table 2 ijms-21-07406-t002:** Strategies to boost ECFCs expansion ex vivo.

Strategy to Improve Regenerative Activity	ECFC Source	Disease or Pathological Condition	Effecton ECFCs	Mechanism of Action	Reference
Acidosis	UCB	Hindlimb ischemia	In vitro proliferation and tubulogenesis, in vivo revascularization	Activates Akt and ERK1/2, inhibits p38	[156]
Acidosis	UCB	T2DM and Hindlimb ischemia	In vitro adhesion and anticytotoxic effect, in vivo revascularization	bFGF, TGFβ1, IL-8, IL-4, VEGF, PDGF, and IL-10	[129]
AA	PB		In vitro proliferation	Activates TRPV4 channel and Ca^2+^-dependent NO release	[167,168]
EPO	UCB		In vitro migration and tube formation	AMPK-Krüppel-like factor 2 (KLF2) and eNOS	[150]
EPO	UCB	Cerebral ischemia	In vivo homing, reduction of BBB disruption, and cerebral apoptosis	CD146 expression	[106]
EPO	UCB	Cerebral ischemia	In vivo angiogenesis and neurogenesis, reduction of infarct volume and neurological deficit	Activation HSP27, STAT-5, Bcl-2, down-regulation of Bax and DP5.Akt-1, BDNF, and VEGF expression	[151]
EPO	UCB	Hindlimb ischemia	In vitro proliferation, migration, tube formation, resistance to H_2_O_2_-induced apoptosis, in vivo revascularization and rescue of blood flow	CD131 and PI3K/Akt	[149]
Fucoidan	UCB	Hindlimb ischemia	In vivo improvement of residual muscle blood flow and increased collateral vessel formation	SDF-1α	[144]
Fucoidan	UCB	Hindlimb ischemia	In vitro rescue from cellular senescence and tube formation, in vivo proliferation, survival, incorporation, and differentiation within neovesselsand recovery of blood flow	FAK, ERK, Akt	[143]
Fucoidan	UCB		In vitro migration	PI3K/Akt	[169]
Genistein	UCB	AMI	In vitro proliferation and migration, in vivo revascularization, improvement of cardiac function and reduction of fibrosis	ILK, α-parvin, F-actin, and ERK1/2	[153]
PL	UCB		In vitro survival, vasculogenesis, and augments blood vessel formation by inhibiting apoptosis	Akt, Bad, and Bcl-xL	[139]
HPL-gel	PB		In vitro proliferation in 2D culture and formation of a complete microvascular network in 3D cultures	VEGF	[141]
Hypoxia	PB	Hindlimb ischemia	In vitro inhibition of cellular senescence, enhances proliferation survival, and angiogenic. In vivo accelerates vascular repair capacity, increases blood flow ratio and capillary density	Hypoxia-inducible factor-1α-TWIST-p21 axis	[158]
Hypoxia	PB	Hindlimb ischemia	In vitro proliferation and survival. In vivo enhanced blood flow ratio, capillary density, and angiogenic cytokine secretion	STAT3–BCL3 axis	[157]
NAADP	PB		In vitro proliferation	Ca^2+^-dependent manner	[170]
Nicotine	UCB		In vitro enhanced viability, adhesion, migration, and tube formation	a7-nAChR	[155]
PRs	PB		In vitro tube formation	Tetraspanin CD151, α6β1 integrin, and Src–PI3K signaling pathway	[171]
SDF-1α	UCB		In vitro enhanced adhesion and migration	SDF-1α and CXCR4	[146]
SDF-1α	UCB	Hindlimb ischemia	In vitro adhesion and tube formation. In vivo accumulation of transplanted ECFCs at sites of ischemia and enhanced neovascularization	MMP-2, FGF-2 CXCR4	[145]
SFLLRN peptide	PB and UCB	Hindlimb ischemia	In vivo increased chemotactic gene expression and leukocyte recruitment at ischemic sites	COX-2 and PAR-1	[53]
SFLLRN peptide	PB and UCB	Hindlimb ischemia	In vitro proliferation	Angiopoietin 2 and PAR-1	[172]

AA: arachidonic acid; AMI: acute myocardial infraction; bFGF: basic fibroblast growth factor; EPO: erythropoietin; HPL-gel: human platelet lysate gel; ILK: integrin-linked kinase; PL: platelet lysate; PRs: platelet releasates; T2DM: type 2 diabetes mellites.

**Table 3 ijms-21-07406-t003:** Strategies to restore angiogenic activity of dysfunctional ECFCs.

Strategy to Improve Regenerative Activity	ECFC Source	Disease or Pathological Condition	Effect on ECFCs	Mechanism of Action	Reference
Adiponectin	PB	T2DM, hindlimb ischemia (normo- and hyperglycemic conditions)	In vitro proliferation and migration and in vivo neovascularization	COX-2	[176]
Acidosis	UCB	T2DM, hindlimb ischemia	In vitro adhesion and release of pro-angiogenic molecules. In vivo increase in capillary density and rescued local blood flow	bFGF, TGFβ1, IL-8, IL-4, VEGF, PDGF, and IL release	[129]
Atorvastatin		H_2_O_2_-induced oxidative damage	In vitro resistance to cell death	Annexin A2 up-regulation	[178]
C1P	UCB	Hindlimb ischemia	In vitro adhesion, proliferation, migration, and tube formation, in vivo vascularization of gel plugs and rescue of blood flow	ERK1/2 and Akt	[109]
FIR	PB of DM and healthy donors	High glucose-induced endothelial dysfunction and hindlimb ischemia	In vitro rescue of migration and tube formation, in vivo revascularization	Down-regulation of miR-134	[177]
FIR	PB of CAD patients and healthy donor	CAD	In vitro rescue of migration and tube formation	Down-regulation of miR-486-5p	[185]
Fucoidan	UCB	Senescence andhindlimb ischemia	In vitro rescue from cellular senescence and tube formation, proliferation, and survival. In vivo incorporation and differentiation within neovesselsand recovery of blood flow	FAK, ERK, and Akt	[143]
Glycomimetic C3	PB	DM with NI or NP foot ulcers, hindlimb ischemia	In vitro rescue of proliferation and tube formation (mainly in NI)	Akt/eNOS and Nrf2/ARE	[173]
Iptakalim	PB		In vitro proliferation and bidimensional tube formation	Akt and eNOS	[179]

ARE: antioxidant response element; C1P: ceramide 1-phosphate; CAD: coronary artery disease; DM: diabetes mellitus; FIR: far-infrared irradiation; IL: interleukin; NI: neuroischemic; NP: neuropathic; Nrf2: nuclear erythroid 2-related factor 2; PB: peripheral blood; T2DM: type 2 diabetes mellitus; UCB: umbilical cord blood.

**Table 4 ijms-21-07406-t004:** Strategies to enhance angiogenic activity of healthy ECFCs.

Strategy to Improve Regenerative Activity	ECFC Source	Disease or Pathological Condition	Effect on ECFCs	Mechanism of Action	Reference
Acidosis	UCB	Hindlimb ischemia	In vitro proliferation and tubulogenesis, in vivo revascularization	Activates Akt and ERK1/2, inhibits p38	[156]
BMP4	UCB and PB	Hindlimb ischemia	In vitro proliferation and tube formation, in vivo revascularization		[189]
EPO	UCB		In vitro migration and tube formation	AMPK-Krüppel-like factor 2 (KLF2) and eNOS	[150]
EPO	UCB	Cerebral ischemia	In vivo homing, reduction of BBB disruption and cerebral apoptosis, rescue of CBF	CD146	[106]
EPO	UCB	Cerebral ischemia	In vivo angiogenesis and neurogenesis, and reduction of infarct volume and neurological deficit	Activation of HSP27, STAT-5, Bcl-2, and down-regulation of Bax and DP5.Akt-1, BDNF, and VEGF expression	[151]
EPO	UCB	Hindlimb ischemia	In vitro proliferation, migration, tube formation, and resistance to H_2_O_2_-induced apoptosis, in vivo neovascularization and rescue of blood flow	CD131 and PI3K/Akt	[149]
Fucoidan	UCB	Hindlimb ischemia	In vivo improvement of residual muscle blood flow and increased collateral vessel formation	SDF-1α	[144]
Fucoidan	UCB	Senescence and hindlimb ischemia	In vitro rescue from cellular senescence and tube formation, proliferation, and survival. In vivo incorporation, differentiation within neovessels, and recovery of blood flow	FAK, ERK, and Akt	[143]
Fucoidan	UCB		In vitro migration	PI3K/Akt	[169]
Genistein	UCB	AMI	In vitro proliferation and migration, in vivo revascularization and improvement of cardiac function, reduction of fibrosis	ILK, α-parvin, F-actin, and ERK1/2	[153]
hAFS-CM	PB	AMI	In vitro bidimensional tube formation and promotion, in vivo revascularization	Through an oscillatory increase in [Ca^2+^]_i_	[190]
OPG	UCB		In vitro migration, chemotaxis, and vascular cord formation. In vivo microvessel formation	SDF-1α, ERK, PI3K/Akt/mTOR	[194]
SDF-1α	UCB		In vitro enhanced adhesion and migration	CXCR4	[146]
SDF-1α	UCB	Hindlimb ischemia	In vitro adhesion and tube formation, in vivo accumulation of transplanted ECFCs at sites of ischemia and enhanced neovascularization	MMP-2, FGF-2 CXCR4	[145]
sCD146		Hindlimb ischemia	In vitro survival and enhanced angiogenesis in vivo	FADD, Bcl-xl, and eNOS	[196]
sCD146	PB from PADpatients	Hindlimb ischemia	In vitro increased clonogenic activity and inhibition of cellular senescence, in vivo neovascularization and recovery of blood flow	miR-21 and embryonic transcription factors	[198]
TSP1	UCB		In vitro migration and tube formation in Matrigel plug, adhesion to an endothelial monolayer	Syndecan 4 and α6-integrin	[193]

BMP4: bone morphogenetic protein 4; hAFS-CM: human amniotic fluid-derived stem cells-conditioned medium; EPO: erythropoietin; OPG: osteoprotegerin; PAD: peripheral artery disease; PB: peripheral blood; TSP1: thrombospondin 1; UCB: umbilical cord blood.

**Table 5 ijms-21-07406-t005:** Strategies to stimulate ECFC homing to sites of neovessel formation.

Strategy to Improve Regenerative Activity	ECFC Source	Disease or Pathological Condition	Effect on ECFCs	Mechanism of Action	Reference
C1P	UCB	Hindlimb ischemia	In vitro adhesion, proliferation, migration, and tube formation, in vivo vascularization and rescue blood flow	ERK1/2 and Akt	[109]
Periostin	UCB	Hindlimb ischemia	In vitro proliferation and adhesion, in vivo homing, blood flow perfusion, and leg salvage	FAS-1, β3, and β5 integrins	[200]
sCD146	UCB	Hindlimb ischemia	In vitro survival and in vivo angiogenesis	FADD, Bcl-xl, and eNOS	[196]
sCD146	PB	Hindlimb ischemia	In vitro migration and tube formation	eNOS, VEGFR2, MMP-2, and uPA	[202]
WKYMVm	UCB	Hindlimb ischemia	In vitro chemotaxis, proliferation, and tube formation. In vivo attenuated tissue necrosis, neovascularization, and recovery of blood flow	FPR2	[199]

C1P: ceramide 1-phosphate; FADD: fas-associated protein with death domain; PAD: peripheral artery disease; PB: peripheral blood; UCB: umbilical cord blood.

**Table 6 ijms-21-07406-t006:** Epigenetic and genetic modulation of ECFCs to rescue their angiogenic activity.

Strategy to Improve Regenerative Activity	ECFC Source	Disease or Pathological Condition	Effect on ECFCs	Mechanism of Action	Reference
Egfl7 silencing	PB	Hindlimb ischemia	In vitro proliferation, differentiation, and migration, in vivo revascularization		[238]
Epigenetic drugs (GSK-343 and Panobinostat, (either alone or in combination)	UCB	Hindlimb ischemia	In vitro migration and tube formation, and resistance to serum starvation-induced apoptosis, in vivo revascularization and rescue of blood flow	VEGFR2, CXCR4, WNT, Notch, and SHH	[207]
FIR	From PB of healthy donor and CAD patients	CAD	In vitro rescue of migration and tube formation	Down-regulation of miR-486-5p	[185]
FIR	From PB of DM and healthy donors	High glucose-induced endothelial dysfunction and hindlimb ischemia	In vitro rescue of migration and tube formation, in vivo revascularization	Down-regulation of miR-134	[177]
FOXF1 overexpression	UCB		Sprouting angiogenesis in vitro and in a zebrafish model	Notch2 and VEGFR2 expression	[235]
NOX4 overexpression	UCB	Hindlimb ischemia	In vitro proliferation, migration, and tubulogenesis. In vivo vasoreparative function and enhanced blood flow	H_2_O_2_ production, PMA-induced superoxide in a NOX4-dependent manner	[233]
Integrin β1 overexpression	PB	Hindlimb ischemia	ECFC homing, in vivo angiogenesis and recovery of blood flow		[230]
Panobinostat	PB	MMD	Restoring capillary-like network formation	Derepressing RALDH	[211]
Akt1 overexpression	PB of South Asian men (at risk for cardiovascular events)	Hindlimb ischemia	In vitro angiogenesis, in vivo re-endothelialization and perfusion recovery	Akt and NO	[229]

CAD: coronary artery disease; DM: diabetes mellitus; FIR: far-infrared irradiation; FOXF1: forkhead box f1; HUVECs: human umbilical vein endothelial cells; MMD: Moyamoya disease; NOX4: NAADP oxidase; PMA: phorbol 12-myristate 13-acetate; RALDH: retinaldehyde dehydrogenase 2.

**Table 7 ijms-21-07406-t007:** Priming ECFCs with mesenchymal progenitor cells (MPCs) and mesenchymal stem cells (MSCs).

Strategy to Improve Regenerative Activity	ECFC Source	Disease or Pathological Condition	Effect on ECFCs	Mechanism of Action	Reference
ECFCs + MPCs	UCB	Hindlimb ischemia	Enhanced blood flow		[57]
ECFCs + MSCs	UCB		Lower risk of ECFC rejection, improved in vivo vascularization	Endothelial HLA-DR expression	[240]
ECFCs + MSCs	UCB	Hindlimb ischemia	In vitro proliferation, migration and tubulogenesis, in vivo enhanced neovessel density	VEGF, HGF and MMP-9	[55]
ECFCs + MSCs	Fetal term placental		In vitro matrigel plugs cell engraftment, and neovascularization	Via direct contact and Notch signaling	[247]
ECFCs + MSCs	UCB	Ischemic gastrocnemius muscle	In vitro proliferation and capillary-like formation, in vivo increase in capillary density	Up-regulating SphK1	[243]
ECFCs + MSCs	Mouse BM		In vitro proliferation, colony formation, migration, and angiogenesis	SDF1-α	[244]

BM; bone marrow; HLA-DR: HLA class II histocompatibility antigen, DR alpha MPCs: mesenchymal progenitor cells; MSCs: mesenchymal stem cells; PB: peripheral blood; UCB: umbilical cord blood.

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
