# Peer review of "Therapeutic Potential of Endothelial Colony-Forming Cells in Ischemic Disease: Strategies to Improve their Regenerative Efficacy"

_ijms, 2020, doi:10.3390/ijms21197406_

Round 1

Reviewer 1 Report

Review, special issue “Endothelial Progenitor Cells in Health and Disease”

Therapeutic Potential of Endothelial Colony Forming Cells in Ischemic Disease: Strategies to Improve Their Regenerative Efficacy

Comments:

The review presented by two of the special issue editors and co-authors is a very elaborate manuscript of 26 pages text and 275 references which by the authors is euphemistically phrased as a brief outline in their abstract. In this brief outline the authors try to cover all 365 degrees of knowledge about ECFCs, including 7 tables.

Although well written it is hard for me to imagine that other contributions are not duplicating this manuscript and makes this manuscript a challenge to read from head to tail. My suggestion for clarity is to condense this manuscript (taking out various details, such as number of cells or pH for ceramide 1-phosphate preconditioning). Furthermore, I would like to be the review more explanatory than enumerating.

To my knowledge in the ECFC field a number of issues still continue not to be present, such as defining the correct ECFC phenotype, their limiting propagation without phenotypic changes, and whether they really form patent vessels in ischemic tissue in human patients. If this would be an introductory review it would be nice to list these (and possible other) issues in the introduction, perhaps with an explanatory visual abstract before addressing these issues and focus on possible therapeutic strategies.

Authors should thoroughly check their references since there are incomplete/incorrect ( 25, 115, 176, 178, 218, 231, 236, 242, 246, 255, 258, 274) and double references (95 = 105; 120 = 58; 121 = 122; 146 = 32).

Author Response

Dear Reviewer #1,
We are gratefully thankful for your comments on our manuscript entitled: “Therapeutic Potential of Endothelial Colony Forming Cells in Ischemic Disease: Strategies to Improve Their Regenerative Efficacy” for publication as Review Article in International Journal of Molecular Sciences – Special Issue Endothelial progenitor cells in health and disease.

The review presented by two of the special issue editors and co-authors is a very elaborate manuscript of 26 pages text and 275 references which by the authors is euphemistically phrased as a brief outline in their abstract. In this brief outline the authors try to cover all 365 degrees of knowledge about ECFCs, including 7 tables.
We thank the Referee for this comment. She/he might have noticed that the expression “brief outline” did not refer to the whole manuscript, but to the short introduction on ECFCs [Paragraph 2.1: Endothelial Colony Forming Cells (ECFCs): Origin, Characterization and Biological Activity]. The manuscript was never meant to be short for the reasons that we are going to illustrate in the next point. We truly thank the Referee for appreciating our efforts to cover all 360 degrees of knowledge about ECFCs.

Although well written it is hard for me to imagine that other contributions are not duplicating this manuscript and makes this manuscript a challenge to read from head to tail. My suggestion for clarity is to condense this manuscript (taking out various details, such as number of cells or pH for ceramide 1-phosphate preconditioning). Furthermore, I would like to be the review more explanatory than enumerating.
We thank the Referee for this comment since she/he raises a philosophical issue. This manuscript was designed at the beginning of the Italian Lockdown, which started on the beginning of March. Apart from the corresponding author, who was busy with remote teaching as his colleagues around the world, the junior investigators of the group had plenty of time to read (and analyse their data). Our laboratory work on signal transduction and, at the present time, we do not have the possibility to investigate the therapeutic efficacy of ECFCs in animal models. We must work in vitro and focus our attention on Ca2+ signalling. Several beautiful and influential reviews have repeatedly addressed the crucial issues regarding ECFC ontology, expansion, and application in regenerative medicine. Many recent reviews (which were mentioned in the text) summarized the main therapeutic applications of ECFCs in cardiovascular diseases by describing the main strategies to improve their therapeutic potential. These are the reviews that we believe the Referee defines as “explanatory” and that we enjoyed a lot: they are extremely informative. On the other hand, we were more interested in the intracellular signalling pathways activated by ECFC manipulation. In our opinion, a comprehensive description of the molecular mechanisms (and their downstream targets) that can be stimulated through pharmacological, genetic, and epigenetic manipulations, was missing. In addition, and this is a personal thought the Referee might not agree with, when I was reading these nice reviews on the vasoreparative potential of ECFCs I was thinking: Maybe they could also mention this treatment, Which is the signalling pathway responsible for this effect, and so on. Therefore, we deliberately chose to write an “enumerating” review. We thought that, while other distinguished Authors had already provided their valuable contribution to summarize the main mechanisms to stimulate ECFCs, a large, comprehensive review to go deeply into the pro-angiogenic signalling pathways was yet to be written. It is unlikely that a reader will go through the review from head to tail and we truly thank the Referees for their careful evaluation of the manuscript. Our desire is that a reader will search for all the information which is currently available on the role of ECFCs in a specific cardiovascular disorder and/or on the pro-angiogenic role of a specific signalling pathway. In the revised version, we removed the pH value for ceramide 1-phosphate preconditioning and the wide description of epigenetic modifications (which were now only enumerated). We understand that the Referee does not agree with our “philosophy”, but we would not like to turn our manuscript upside-down since we were fully aware of what we were writing. While the existing reviews in the field are certainly “explanatory” and summarize the main findings in the field, we adopted a more encyclopaedic approach.

To my knowledge in the ECFC field a number of issues still continue not to be present, such as defining the correct ECFC phenotype, their limiting propagation without phenotypic changes, and whether they really form patent vessels in ischemic tissue in human patients. If this would be an introductory review it would be nice to list these (and possible other) issues in the introduction, perhaps with an explanatory visual abstract before addressing these issues and focus on possible therapeutic strategies.
We thank the Referee for this comment. We addressed these issues in Paragraph 2.1: Endothelial Colony Forming Cells (ECFCs): Origin, Characterization and Biological Activity. A recent consensus statement on nomenclature, presented by the major experts in the fields, identified ECFCs as truly endothelial progenitors, which display high clonogenic potential, truly belong to the endothelial lineage, assemble into capillary-like networks in Matrigel scaffolds in vitro and, in vivo, form functional endothelial cell vessels which anastomose with the host vasculature. Actually, Mervin C. Yoder already wrote several, wonderful reviews to cover these aspects of ECFC physiology (PMID: 28915234; PMID: 28296182; PMID: 23794280; PMID: 22762017). All these details were already mentioned in the Introduction (Lines 111-134 of the revised manuscript). In addition, we now described the genomic stability of ECFCs in culture (Lines 134-136) and added a visual abstract to the text.

Authors should thoroughly check their references since there are incomplete/incorrect ( 25, 115, 176, 178, 218, 231, 236, 242, 246, 255, 258, 274) and double references (95 = 105; 120 = 58; 121 = 122; 146 = 32).
We thank the referee for this comment. We carefully went through the references and corrected all the incomplete ones as well as we eliminated all the duplicates.

Once again we truly thank you for the careful evaluation of our manuscript and do hope that you will now regard our manuscript suitable for publication on the International Journal of Molecular Sciences.

Reviewer 2 Report

I would like to congratulate the authors on writing a very extensive review of endothelial progenitor cells, focusing on ECFC and their regenerative potential in ischemic disease. In my opinion, this review is an excellent summary of the current literature, and the authors have done well to define a very complicated and, often, controversial field of research.

There are many published reviews of ECFC- often focusing on their origin, phenotype and pre-clinical studies of their capacity for vascular regeneration. Whilst these topics are, rightly-so, included in this review, it also contains novel information, with a focus on strategies to improve the regenerative potential of ECFC e.g. using pharmacological agents, MSC priming or genetic manipulation. ECFC have high clinical potential for regeneration of ischemic tissues, and therefore, the translation focus of this review is important.

Comments:

  • Unfortunately, the review requires quite extensive improvement of the written English, as there are many spelling and grammatical errors throughout.
  • My other main concern is that the Abstract is cut and pasted from the main body of the text, and should be improved by using original text.

I have a few minor comments:

Line 120:

“Nevertheless, a recent investigation by Fujisama and coworkers demonstrated that ECFCs isolated 120 from peripheral blood and arterial wall of male donors previously transplanted with female bone 121 marrow, exhibited an XY genotype [34]. “

The first author of this study is Fujisawa, and the ECFCs were isolated from the venous and not arterial wall. Male ‘patients’ should be used rather than donors to avoid confusion since these patients received female donor bone marrow.

  • Line 126:

“stem cell niches” seems inaccurate, as stem and progenitor cells are different. I advise removing ‘stem cells’ therefore using the more accurate ‘niches’ or ‘progenitor cell niches’.

  • A reference is required to support the statement on line 253

Author Response

Dear Reviewer #2,
We are gratefully thankful for your comments on our manuscript entitled: “Therapeutic Potential of Endothelial Colony Forming Cells in Ischemic Disease: Strategies to Improve Their Regenerative Efficacy” for publication as Review Article in the International Journal of Molecular Sciences – Special Issue Endothelial progenitor cells in health and disease.

Major Comments:
Unfortunately, the review requires quite extensive improvement of the written English, as there are many spelling and grammatical errors throughout.
The Referee is right. The manuscript has been extensively edited and the spelling and grammatic errors were amended.

My other main concern is that the Abstract is cut and pasted from the main body of the text, and should be improved by using original text.
The Referee is again right. The Abstract has been rewritten by using original text.

Minor comments:
Line 120: “Nevertheless, a recent investigation by Fujisama and coworkers demonstrated that ECFCs isolated 120 from peripheral blood and arterial wall of male donors previously transplanted with female bone 121 marrow, exhibited an XY genotype [34]. “ The first author of this study is Fujisawa, and the ECFCs were isolated from the venous and not arterial wall. Male ‘patients’ should be used rather than donors to avoid confusion since these patients received female donor bone marrow.
The Referee is right, and we are grateful for the attention she/he paid to these details. The manuscript has been amended to address these criticisms.

Line 126: “stem cell niches” seems inaccurate, as stem and progenitor cells are different. I advise removing ‘stem cells’ therefore using the more accurate ‘niches’ or ‘progenitor cell niches’.
We thank the referee for this comment and the manuscript has been revised accordingly.

A reference is required to support the statement on line 253.
A reference has been added, as rightly requested by the Referee.